# Effects of Ammonia and Salinity Stress on Non-Volatile and Volatile Compounds of Ivory Shell (*Babylonia areolata*)

**DOI:** 10.3390/foods12173200

**Published:** 2023-08-25

**Authors:** Chunsheng Liu, Yunchao Sun, Xin Hong, Feng Yu, Yi Yang, Aimin Wang, Zhifeng Gu

**Affiliations:** 1Sanya Nanfan Research Institute, Hainan University, Sanya 572022, China; fengyu2022@hainanu.edu.cn (F.Y.); yiyangouc@outlook.com (Y.Y.); hnugu@163.com (Z.G.); 2College of Marine Biology and Fisheries, Hainan University, Haikou 570228, China; yiyyiziwei1@outlook.com (Y.S.); hongx0866@163.com (X.H.); aimwang@163.com (A.W.)

**Keywords:** *Babylonia areolata*, non-volatile compounds, volatile compounds, environmental stress

## Abstract

In this study, the flavor compounds of ivory shell (*Babylonia areolata*) and their changes caused by ammonia and salinity stresses were studied. Ammonia stress improved the contents of free amino acids (FAAs), 5′-adenosine monophosphate (AMP), citric acid, and some mineral ions such as Na^+^, PO_4_^3−^, and Cl^−^. The FAA contents decreased with increasing salinity, while the opposite results were observed in most inorganic ions (e.g., K^+^, Na^+^, Mg^2+^, Mn^2+^, PO_4_^3−^, and Cl^−^). Hyposaline and hypersaline stresses increased the AMP and citric acid contents compared to the control group. The equivalent umami concentration (EUC) values were 3.53–5.14 g monosodium glutamate (MSG)/100 g of wet weight, and the differences in EUC values among treatments were mainly caused by AMP. Hexanal, butanoic acid, and 4-(dimethylamino)-3-hydroxy- and (E, E)-3,5-octadien-2-one were the top three volatile compounds, and their profiles were significantly affected when ivory shells were cultured under different ammonia and salinity conditions.

## 1. Introduction

The ivory shell (*Babylonia areolata*) is an aquatic animal widely distributed across tropical and subtropical coastal areas of the Indo-Pacific Ocean [1,2]. Recently, ivory shell culture has expanded rapidly in many East and Southeast Asia countries because of its faster growth rate, higher edible soft tissue ratio, and higher tolerance to unsuitable environmental conditions [2,3,4]. In China, the annual production of this species exceeded 20,000 tons, in which Hainan, Guangdong, Fujian, and Guangxi Provinces were the main cultural areas [5]. Currently, ivory shells are cultured in semi-outdoor flow-through systems and fed with forage fish (such as *Decapterus maruadsi*), and their breeding density is approximately 5–8 kg per square meter [3,4]. However, this type of low-tech breeding model is often affected by the fluctuation of various environmental factors, such as temperature, ammonia, and salinity, causing low growth rate and even death [2,4,6].

Recently, studies on the adversely effects of elevated ammonia and unfavorable salinity levels on ivory shells have been undertaken. Zhou et al. [2] showed that higher ammonia (≥5.0 mg/L) significantly depressed the ivory shells’ growth performance, and the amino acid and fatty acid contents were significantly affected after long-time culture. Liu et al. [4] reported that ivory shells cultured under higher-salinity conditions (35 and 40 ppt) showed obvious changes in the stress-related enzymes activities and body biochemical compositions compared to lower-salinity (20–30 ppt) groups. During their life, aquatic animals suffer from environmental variation and stress due to variables such as temperature, salinity, pH, oxygen, and ammonia [7]. To reduce and eliminate these adverse effects, the cytoprotective and detoxification pathways of aquatic animals could be enhanced, causing higher energy consumption and lower growth performance [8]. Nutritional composition (lipid, amino acid, and glucose) and energy metabolism changes caused by these environmental stresses have also been reported in many aquatic animals, such as ivory shell, razor clam (*Sinonovacula constricta*), Pacific white shrimp (*Litopenaeus vannamei*), and Chinese mitten crab (*Eriocheir sinensis*) [2,4,8,9,10,11].

In addition to the above major essential nutrition, seafood contains several pleasant non-volatile compounds (e.g., free amino acids (FAAs) and flavor nucleotides, organic acids, and inorganic ions) and volatile organic compounds, including aliphatic hydrocarbons, alcohols, aldehydes, esters, and ketones [12,13,14,15]. These non-volatile and volatile compounds contribute to the flavor of seafood [12]. Ammonia and salinity are two important factors that influence the flavor compounds of aquatic animals [11,13]. In previous studies, environmental changes were shown to have an impact on the physiological metabolism (such as glycolysis, lipolysis, respiratory metabolism, etc.) of aquatic animals, leading to changes in non-volatile and volatile compounds [11,13,16]. For example, razor clams cultured at high salinity (20–25 ppt) possessed more positive, abundant volatiles than those cultured at low salinity [9]; low salinity decreased the content of flavor compounds and weakened the umami taste and aroma characteristics of mud crab (*Scylla paramamosain*) [17]; high ammonia concentrations could reduce umami amino acids and increase bitter amino acid levels in Chinese mitten crab meat [11]; freshwater fish (such as catfish) had stronger muddy odor compared to marine fish, which was mainly caused by higher contents of 2-methylisoborneol, o (E)-2-nonenal, 1-octen-3-ol, and other earthy-musty compounds in freshwater fish [18,19].

Besides providing delicious taste, the change of flavor compounds of aquatic animals is also one important physiological adaptation mechanism to environmental stress. FAAs and nucleotides are important bioactive components and play an important physiological function for resisting environmental stress [20]. FAAs and inorganic ions can regulate osmotic pressure [21]. Organic acids can improve antibacterial activity and inhibit necrosis and apoptosis caused by cardiomyocyte hypoxia/reoxygenation [22]. Volatile compounds such as aldehydes, ketones, and alcohols are closely related to the oxidative metabolism of proteins and lipids [13].

Our previous studies have shown that the body biochemical compositions of ivory shells are significantly changed by ammonia and salinity [2,4]. However, there are no reports on detecting non-volatile and volatile compounds in this species and on the effects of ammonia and salinity stresses on the changes of flavor compounds. Therefore, the influence of five different ammonia and salinity levels on the contents of non-volatile compounds (e.g., FAAs, 5′-nucleotides, organic acids, and inorganic ions) and volatile organic compounds of the juvenile ivory shell was investigated after an eight-week rearing period. The findings of this study can provide insights into the management of ivory shell culture environment and improve its meat taste.

## 2. Materials and Methods

### 2.1. Experimental Animals

Ivory shells (body weight 2.50 ± 0.17 g) were purchased from a local aquatic farm and acclimatized in the laboratory for 14 days before the experiment. Ivory shells were maintained in at ~28 °C in aerated seawater (pH (7.7 ± 0.4) and salinity (31.2 ± 0.8 ppt)) and fed with forage fish daily (~4% of body weight) at 3:00 in the afternoon, and 3/4 of the water volume was changed 2 h after the feeding. 

### 2.2. Experimental Design and Sampling

Based on our previous studies [2,4], five ammonia concentrations (0, 2.5, 5.0, 7.5, and 10.0 mg/L) and salinity levels (20, 25, 30, 35, and 40 ppt) of long-term stress were set. In ammonia stress experiments, ammonium chloride (NH_4_Cl, Sinopharm Chemical Reagent Co., Ltd., Shanghai, China) was added to achieve the desired final ammonia concentration. All ammonia-treated groups were conducted in triplicate. For each group, a total of 90 ivory shells were randomly divided into three sub-groups and cultured in three 68 L blue tanks (30 ivory shells for each tank). The cultured condition of each ammonia stress group was the same as temporary rearing experiment except that the water was changed with the same ammonia concentration before. For salinity stress experiments, the experimental condition was as same as above, and the water salinity of each salinity-treated group was maintained by adding commercial sea salt or purified freshwater to the natural seawater (~31.2 ppt). 

Ivory shells were sampled after eight weeks of rearing. In detail, the ivory shells were sacrificed 24 h after the last feeding. The muscle of ivory shells in each tank was sampled, quickly frozen with liquid nitrogen, and stored at −80 °C for non-volatile and volatile compounds analyses. All chemical analyses were performed within two months.

### 2.3. FAA Analysis

FAAs were extracted using a method previously reported by Liu et al. [14] and Sun et al. [15]. Ivory shell muscle (2.5 g) was homogenized with 7.5 mL of 10% trichloroacetic acid (TCA, Sinopharm Chemical Reagent Co., Ltd., Shanghai, China). Supernatants were centrifuged at 10,000× *g* for 15 min (4 °C) and separated into 25 μL aliquots. Samples were analyzed by high-performance liquid chromatography (HPLC) in a Waters 2996 (Waters Corporation, Milford, MA, USA). In detail, samples (5 µL) were separated using a Waters Pico-Tag-C18 column (3.9 mm × 150 mm). The mobile phase consisted of AccQ-Tag Elent A (*v*/*v* = 10:1) (A), 100% acetonitrile (B), and deionized water (C). Gradient elution was performed as follows: 100% A to 99% A and 1% B at 0.5 min, to 94% A and 6% B at 18 min, to 90% A and 10% B at 19 min, to 83% A and 17% B at 29.5 min, to 60% B and 40% C at 33 min, to 100% A at 36 min, and to 100% A at 45 min. The detection wavelength was 248 nm. All analyses were repeated in triplicate.

### 2.4. 5′-Nucleotide Analysis

The 5′-nucleotides in the ivory shell were performed using a previously described method [23]. HPLC conditions for 5′-nucleotide assays were as follows: injection volume (20 µL); column temperature (30 °C); eluent A (methanol, Sinopharm Chemical Reagent Co., Ltd., Shanghai, China) and eluent B (0.05% phosphoric acid, Sinopharm Chemical Reagent Co., Ltd., Shanghai, China). The following gradient was performed: initial of 5% A for 10 min, linear change to 15% A for 5 min, linear change to 70% A for 6 min, and finally, linear change to 5% A for 4 min; flow rate (1.0 mL/min); detection wavelength (260 nm). All analyses were repeated in triplicate.

### 2.5. Equivalent Umami Concentration (EUC)

The EUC (g monosodium glutamate (MSG)/100 g sample weight) is the concentration of MSG equivalent to the umami intensity determined by the mixture of umami amino acids and 5′-nucleotides and is represented by the following equation:Y = Σ*a*_i_*b*_i_ + 1218(Σ*a*_i_*b*_i_) (Σ*a*_j_*b*_j_)
where a_i_ is the concentration (g/100 g) of respective umami amino acid (glutamic acid (Glu) and aspartic acid (Asp); b_i_ is the relative umami concentration (RUC) for umami amino acids to MSG (Glu and Asp were 1 and 0.077, respectively); a_j_ is the concentration (g/100 g) of 5′-nucleotide (5′-inosine monophosphate (IMP), 5′-guanosine monophosphate (GMP), and 5′-adenosine monophosphate (AMP)); b_j_ is the RUC for umami 5′-nucleotides to IMP (IMP, GMP, and AMP were 1, 2.3, and 0.18, respectively) [14].

### 2.6. Organic Acid Analysis

The lactic, acetic, citric, malic, and succinic acid contents were extracted and analyzed as described by Liu et al. [23]. Ivory shell muscle (2 g) was homogenized in 10 mL of purified water for 5 min. The supernatants were centrifuged at 10,000× *g* (4 °C) for 20 min and filtered through a 0.45 μm cellulose membrane (Whatman GmbH, Dassel, Germany) before HPLC analysis. The HPLC conditions were almost the same as the 5′-nucleotide analysis in Section 2.4 except that the detection wavelength was set at 215 nm. All analyses were repeated in triplicate.

### 2.7. Inorganic Ion Analysis 

Based on our previous study [23], the concentrations of six mental ion compositions (Ca^2+^, Na^+^, K^+^, Mg^2+^, Zn^2+^, and Mn^2+^) in ivory shells were analyzed using flame atomic absorption spectrophotometry (AA-6800, Shimadzu Corporation, Tokyo, Japan). The concentrations of Cl^−^ and PO_4_^3−^ were analyzed by ion chromatography with a Metrohm model 882 ion chromatograph (Metrohm Ltd., Herisau, Switzerland) according to the method described by Liu et al. [23]. All analyses were repeated in triplicate.

### 2.8. Volatile Compounds Analysis

The volatile compounds of the ivory shell muscles were detected by headspace monolithic material sorptive extraction–gas chromatography–mass spectrometry (HS-MMSE-GC-MS). The sample (2 g), NaCl (1.125 g, Sinopharm Chemical Reagent Co., Ltd., Shanghai, China), and a magnetic stirrer were placed in a glass vial and heated at 70 °C for 40 min. The volatile compounds were absorbed by a solid-phase microextraction (SPME) apparatus (DVB/CAR/PDMS, Supelco Inc., Bellefonte, PA, USA) in the headspace of the vial. The compounds concentrated in the SPME apparatus were desorbed in a thermal desorption unit (TDU, Gerstel GmbH & Co. KG, Mülheim an der Ruhr, Germany) and injected into a GC-MS (7890A-5975C; Agilent, Santa Clara, CA, USA) via a cooled injection system (CIS, Gerstel, Germany). The GC and MS parameters were as follows: column, HP-5MS (30 m × 250 μm × 0.25 μm, Agilent); carrier gas, helium (99.999%); flow rate, 3.0 mL/min; detector interface temperature, 280 °C; ion source temperature, 230 °C; ionization energy, 70 eV. The oven temperature ranged from 40 °C to 160 °C at 3 °C/min, then to 230 °C at 15 °C/min, and held for 2 min. Mass spectra were scanned from 50 to 550 amu in total ion chromatogram (TIC) mode to identify the various compounds. All analyses were repeated in triplicate.

### 2.9. Statistical Analysis

Values are expressed as the mean ± standard deviation (S.D.). Volatile and non-volatile compounds data of different ammonia and salinity groups were compared using one-way analysis of variance (ANOVA), followed by the least significant difference (LSD) test at the significance level of 0.05 (*p* < 0.05). All statistical analyses were conducted using the SPSS 15.0 software (SPSS Inc., Chicago, IL, USA).

## 3. Results and Discussion

### 3.1. Comparison of FAA and Flavor 5′-Nucleotide Compositions in Ivory Shells

As shown in Table 1, twenty FAAs were identified in ivory shells. Of these, glycine (Gly) was the most abundant (204.05–224.43 mg per 100 g of wet weight), followed by glutamic acid (Glu, 111.34–122.11 mg per 100 g of wet weight) and alanine (Ala, 79.45–87.39 mg per 100 g of wet weight). In previous reports, the main FAAs of marine aquatic animals were various. For example, the top three FAAs were Ala, Gly, and Glu in Hongkong oyster (*Crassostrea hongkongensis*), which were almost the same as those in ivory shell [24], while Gly, Ala, and histidine (His) were prevalent in Suminoe oyster (*Crassostrea ariakensis*) [14]; for mud crab, the main FAAs were proline (Pro), arginine (Arg), and Gly [25]; for Pacific white shrimp, they were lysine (Lys), leucine (Leu), and Arg [26]; and for grouper, they were Gly, Ala, and Lys [15]. 

As for the total FAAs, the values of ivory shells were 7.30–8.03 mg/g of wet weight, which were almost the same as Hongkong oyster (7.76 mg/g of wet weight), lower than crab (18.93–36.94 mg/g of wet weight), and higher than Suminoe oyster and groupers (6.79 and ~2.05 mg/g of wet weight, respectively) [14,24,25]. Different FAAs can contribute to various tastes. For example, aspartic acid (Asp) and Glu can provide an umami taste. Gly, Ala, Arg, and the other three FAAs contribute to a sweet taste, while His, Lys, Leu, and the other seven FAAs provide a negative, bitter taste [14,25]. The contents of MSG-like and sweet FAAs in ivory shells were 158.56–174.38 and 426.10–468.62 mg per 100 g of wet weight, respectively, which were significantly higher than bitter FAA contents (84.47–92.11 mg per 100 g of wet weight). According to the taste threshold of each FAA reported by Liu et al. [14,25], the contents of Glu, Gly, Arg, and Ala in ivory shells were higher than their taste activity values (TAVs > 1), which were considered as active FAAs. These results indicated that ivory shells had excellent FAA composition.

The FAA contents of ivory shells significantly increased with increased ammonia concentration, while the values significantly decreased with increasing salinity (*p* < 0.05). FAAs regulate intracellular osmolarity and detoxification function when suffering from various environmental stressors in aquatic animals [27,28]. Several studies on alternations of FAAs in the different tissues have been performed on fish, shrimp, crab, and oyster cultured in different ammonia or salinity conditions [27,28,29]. Under ammonia stress conditions, aquatic animals reduce their feed intake. However, the lower food supply cannot meet the energy requirements of basal physiological metabolism [28]. Some FAAs such as Arg, Glu, and His can be converted into alanine and α-ketoglutarate by alanine aminotransferase, providing extra energy to resist ammonia stress [30]. Therefore, more FAAs were observed under higher ammonia concentration in ivory shells. Correspondingly, the total amino acid compositions significantly decreased with an increase in ammonia concentration in our previous study [2]. As for the relationship between FAA contents and salinity changes, Song et al. [27] observed the significantly increasing tendency of oysters’ total FAAs when transferred to hyper-salinity. Yang et al. [29] reported that the highest concentration of Chinese mitten crab FAAs (such as Asp, Glu, Ala, and Arg) was detected in the low-salinity group (8 ppt). As for ivory shell, the lower-salinity groups (20 and 25 ppt) had the highest FAA contents, which showed the same tendency as Chinese mitten crab [29]. These results indicate that FAAs lead to osmotic regulation in low salinity. Additionally, the FAA contents also showed an opposite tendency with total amino acid compositions in ivory shell muscle, which may be related to amino acid metabolism and energy-expensive physiological mechanisms [4].

Table 2 shows the concentrations of nucleotides in the ivory shell meat under different ammonia and salinity levels. Only two 5′-nucleotide (AMP and UMP) were detected, with AMP being the main nucleotide component (129.90–180.96 mg/100 g of wet weight). The contents of AMP significantly increased with an increase in ammonia concentration. The highest AMP content was observed in the group treated with 25 ppt, which showed significantly higher contents than that in the group treated with 30 ppt (*p* < 0.05). AMP is one of the most important components of umami taste, which promotes sustainable, complex, and umami taste and sweetness in seafood [12,25]. Previous studies reported that the content of AMP in oyster (*C. gigas*, *C. ariakensis*) and squid (*Todarodes pacificus*) was the highest, which showed the same tendency as ivory shell [12,14,31], while in Yangtze (*Coilia ectenes*), the highest 5′-nucleotide was IMP content [32]. Besides providing an umami taste, nucleotides, known as antioxidative materials, could modulate immune responses and enhance stress tolerance in many aquatic animals [33]. In ivory shell, ammonia, hyposaline, and hypersaline stress could increase the contents of AMP. These results indicate that AMP plays an important role in resistance to ammonia and salinity stress. 

### 3.2. Comparison of Organic Acid and Inorganic Ions in Ivory Shells

As shown in Table 3, lactic, acetic, and citric acids were detected in ivory shell, and their contents in muscle were 2.06–3.74, 2.63–5.43, and 26.07–39.00 mg/g of wet weight, respectively. These results indicate citric acid is a major acidic component responsible for the taste of ivory shell. The organic acids detected in seafoods were mainly succinic, citric, malic, lactic, acetic, and propionic acids [25]. These compounds were significantly different in composition and concentrations according to species and their culture conditions [34]. In Suminoe oyster, citric and succinic acids were the two detected organic acids [14]. In squid, the succinic and lactic acids were detected [12]. In mud crab, succinic, citric, and lactic acids were found [25]. Besides the acidic taste, citric acid could contribute to the mollusk’s soft, crisp, acidic taste [14].

The citric acid content of ivory shell significantly increased with an increase in ammonia concentrations (*p* < 0.05). The content of citric acid first decreased and then increased with increasing salinity, and its lowest value was observed in 30 ppt. It has been reported that the sample’s freshness can also alter the content of organic acids by glycolysis [35]. In this experiment, the ivory shell muscles were well preserved (stored at −80 °C for no more than two months). Therefore, the differences of organic acid contents of different ivory shell groups were not caused by storage condition. Furthermore, organic acids are closely related to the synthesis and metabolism of aromatic compounds, amino acids, and esters [24]. Citric acid and other organic acids are the key intermediary metabolism substance in the tricarboxylic acid cycle (TAC), which can provide nutrition and energy for animals to resist environmental stress [36]. Therefore, relatively higher citric acid contents were observed at ammonia and salinity stress.

The contents of Ca^2+^, K^+^, Na^+^, Mg^2+^, Zn^2+^, Mn^2+^, PO_4_^3−^, and Cl^−^ ions in ivory shell muscle with different ammonia- and salinity-treated groups are listed in Table 4. Of these eight inorganic ions, the contents of Na^+^ were the highest, followed by Mg^2+^, K^+^, Ca^2+^, PO_4_^3−^, and Cl^−^ and two trace elements, namely Zn^2+^ and Mn^2+^, at natural conditions. When suffering from ammonia stress, the contents of Na^+^, PO_4_^3−^, and Cl^−^ in ivory shell significantly increased with increasing ammonia concentration (*p* < 0.05), while the Mg^2+^ contents showed a decreasing tendency. Ca^2+^, Zn^2+^, and Mn^2+^ contents first increased and then decreased, and the K^+^ contents were almost the same. As for the salinity stress, most inorganic ions (e.g., K^+^, Na^+^, Mg^2+^, Mn^2+^, PO_4_^3−^, and Cl^−^) showed an increasing tendency with increasing salinity. 

In food products, inorganic ions mainly give bitter and salty tastes, but they could assist in the formation of other flavors. Na^+^ and Cl^−^ could improve the umami taste of FAAs and 5′-nucleotides [12]. PO_4_^3−^ could increase the intensities of MSG-like tastes and sourness and reduce bitterness [14]. Cl^−^ could suppress the sour taste of food [12]. Zn^2+^ is an essential mineral, and its RDA is 3–13 mg/day [37]. The variation in the mineral composition of seafoods could be attributed to species, harvest season, and cultural condition [22,37]. When comparing the inorganic ions of the ivory shell with other mollusk species, the contents of Ca^2+^ and Mg^2+^ in the ivory shell were higher than the Hongkong oyster, Suminoe oyster, and Asian hard clam (*Meretrix lusoria*) [14,24,37]. K^+^ and Na^+^ contents were lower than Hongkong oyster and higher than Suminoe oyster and Asian hard clam [14,24,37]. PO_4_^3−^ and Cl^−^ contents were almost the same as in Hongkong oyster and lower than in Suminoe oyster. Ca^2+^ is responsible for shell development and maintenance [14,24]. Mg^2+^ is a co-enzyme and essential to many biochemical reactions in organisms. The higher Ca^2+^ and Mg^2+^ contents in ivory shell compared to other mollusk species might be ascribed to the rapid growth rate of this species [14,24,37]. 

Inorganic ions are important in regulating aquatic animals’ vital physiological and biochemical functions and maintaining their normal life processes [38]. K^+^, Na^+^, and Cl^−^ are electrolytes that can regulate cell osmotic pressure; Ca^2+^ and PO_4_^3−^ are involved in the development and maintenance of the skeletal system and perform many other biochemical reactions; Mg^2+^, Zn^2+^, and Mn^2+^, as part of enzymes or as a co-factor, play an essential physiological role in many functions in the body [39,40]. Our study shows that ammonia and salinity stress break the physiological balance of ivory shell, leading to changes in these minerals.

### 3.3. Effect of Ammonia and Salinity on EUC of Ivory Shells

The impact of ammonia and salinity stress on the EUC of ivory shells is shown in Figure 1. The EUC values were 3.53–5.14 g MSG/100 g of wet weight, which were lower than in oysters (6.47 and 8.80 g MSG/100 g for Hongkong oyster and Suminoe oyster, respectively) and higher than in squids (0.25 g MSG/kg) [12,14,24]. The EUC values increased with an increase in ammonia concentration, and significant differences were observed in 2.5 and 10 mg/L ammonia-treated groups vs. the control group (*p* < 0.05). Similarly, hyposaline and hypersaline stress also caused an increase in EUC. The EUC values were calculated using MSG-like amino acids (Glu and Asp) and 5′-nucleotides (IMP, GMP, and AMP). Therefore, the differences in EUC values of different treatments were mainly caused by changes in AMP content.

Besides umami amino acids and 5′-nucleotides, many other flavor compounds are related to umami taste. Na^+^, Cl^−^, and PO_4_^3−^ can enhance the overall umami taste, while Ca^2+^ negatively correlated with EUC [12,24]. Based on the changes of these inorganic ions, the umami taste in higher ammonia- and salinity-treated ivory shell muscle was increased because higher Na^+^, Cl^−^, and PO_4_^3−^ contents were detected in these groups. Until now, the EUC is still an important indicator in the evaluation of the umami taste of many types of seafood, such as oysters, crabs, and fish [14,15,24,25]. In ivory shells, Glu and AMP were the main flavor compositions, which showed the same results as oysters (*C. ariakensis*) [14].

### 3.4. Changes in Volatile Components in Ivory Shells

Thirty-nine volatile compounds were detected in the muscles of ivory shell, including 6 aliphatic hydrocarbons, 1 alcohol, 16 aldehydes, 13 ketones, 2 acids, and 1 sulfocompound (Appendix A). The dominant volatiles were aldehydes (47.10–56.23%), followed by ketones, acids, and aliphatic hydrocarbons. In previous reports, cooked razor clam shared the same tendency as ivory shells, while in both raw and boiled Asian hard clams, alcohol was the most abundant volatile compound [9,40]. Aldehydes had low aroma-threshold values, which had stronger positive effects on the marine aroma compared to alcohol [41]. Ivory shell probably presents more abundant flavors because of higher profiles in these odor-active compounds. 

The factors affecting volatile compounds include species, diet, breed, and processing [42]. In the ivory shell, hexanal, butanoic acid, and 4-(dimethylamino)-3-hydroxy- and (E, E)-3,5-octadien-2-one were the top three volatile compounds, and their profiles were 22.71–32.39%, 6.64–14.60%, and 8.85–11.24%, respectively. According to previous studies, the major volatile compounds of mollusk species were ethanol dimer, propan-1-ol, and furan-3-ylmethanol in Asian hard clam [40]; pentanal, 1-pentanol, and hexanal in cooked razor clam [9]; and 3-octene, 2,2-dimethyl-, 3-heptene, 4-propyl-, and pentadecane in squid [12]. Furthermore, various changes in these volatile compounds caused by ammonia and salinity stresses were observed. For example, hexanal showed an upward trend with an increase in ammonia concentration, and its value first increased and then decreased with an increasing salinity from 20 to 40 ppt; for butanoic acid, i.e., 4-(dimethylamino)-3-hydroxy-, the opposite results were observed when compared to hexanal; for (E,E)-3,5-octadien-2-one, there were almost no significant differences among different treatments. Similarly, the changes in the volatile content occurred when the culture salinity of razor clam was raised [9].

Production of volatiles in seafood depends on lipid auto-oxidation and enzymatic oxidation [40]. Aldehydes are mainly formed from lipid oxidation, and most of them have a fatty and fruity aroma [43]. As the major aldehyde, hexanal is a derivative of linoleic acid and arachidonic acid oxidation, which has an odor described as oxidized, fatty, green, grassy, powerful, and penetrating [9]. In our previous study, only linoleic acid was detected in the ivory shell, and its profile in the total lipid of muscle showed the opposite tendency with hexanal when the ivory shell suffered the same ammonia and salinity stresses [2,4]. We indicated that hexanal in ivory shell was probably produced from linoleic acid oxidation. Acids are formed through the oxidation of aldehydes and have cheesy, ammoniacal odors [44]. In ivory shell, as the butanoic acid, i.e., 4-(dimethylamino)-3-hydroxy-, accounted for 6.64–14.60% of total detected volatile components, this acid might have a great effect on the favor of ivory shell. Ketones are produced via microbially induced oxidation, lipid oxidation, or amino acid degradation and possess a heavy rose fragrance [45]. In the ivory shell, (E, E)-3,5-octadien-2-one is the major ketone. (E, E)-3,5-octadien-2-one can provide a sweet, milky odor in aquatic animals, and its relatively high intensity has been reported in steamed Chinese mitten crab [46]. Therefore, the high content of (E, E)-3,5-octadien-2-one might contribute a positive odor to the flavor of the ivory shell.

## 4. Conclusions

The non-volatile and volatile compounds of ivory shell and their changes caused by ammonia and salinity stresses were first demonstrated in this study. The contents of umami and sweet FAAs were significantly higher than bitter FAAs, and their contents showed an increasing tendency with increasing ammonia concentration, while the opposite tendency was observed in the salinity stress experiment. AMP and citric acid were the major 5′-nucleotide and organic acid, respectively, and ammonia, hyposaline, and hypersaline stresses increased their contents compared to the control group. The EUC values of ivory shell were 3.53–5.14 g MSG/100 g of wet weight, and the differences among treatments were mainly caused by AMP. When suffering from ammonia stress, the contents of Na^+^, PO_4_^3−^, and Cl^−^ in ivory shell significantly increased with increased ammonia concentration, while the Mg^2+^ contents showed decreasing tendency. Ca^2+^, Zn^2+^, and Mn^2+^ contents first increased and then decreased, and the K^+^ contents were almost the same. As for salinity stress, most inorganic ions (such as K^+^, Na^+^, Mg^2+^, Mn^2+^, PO_4_^3−^, and Cl^−^) showed an increasing tendency with increasing salinity. Hexanal, butanoic acid, and 4-(dimethylamino)-3-hydroxy- and (E, E)-3,5-octadien-2-one were the top three volatile compounds, and their profiles were significantly affected when ivory shells were cultured under different ammonia and salinity conditions. In addition to non-volatile and volatile compounds, other quality indexes such as texture, water-holding capacity, color, cooking loss, and post-harvest flavor changes of the muscle should be detected in follow-up study. In conclusion, slight ammonia elevation (2.5 mg/L) and low salinity (25 ppt) could improve the characteristic taste of ivory shell.

## Figures and Tables

**Figure 1 foods-12-03200-f001:**
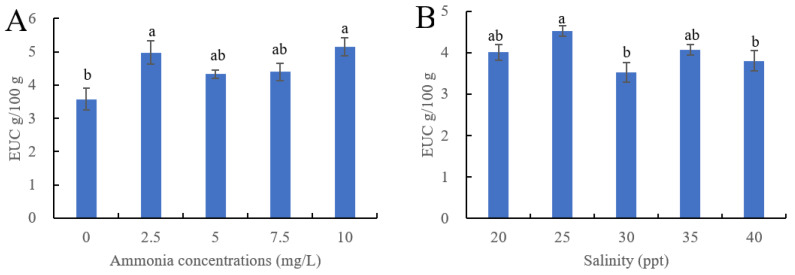
The EUC of ivory shell meat cultured under different ammonia (**A**) and salinity levels (**B**). Bars with different letters were significantly different (*p* < 0.05).

**Table 1 foods-12-03200-t001:** The contents (mg/100 g wet weight) and taste attributes (+pleasant, −unpleasant) of free amino acids (FAAs) in the ivory shell meat under different ammonia and salinity levels.

Item	Taste Attribute	Ammonia Concentrations (mg/L)					Salinity (ppt)				
		0	2.5	5	7.5	10	20	25	30	35	40
Aspartic acid ^1^	Umami (+)	49.77 ± 0.48d	50.25 ± 0.36cd	50.80 ± 0.56bc	51.28 ± 0.41ab	51.93 ± 0.05a	51.78 ± 0.18A	51.66 ± 0.09A	47.76 ± 0.05B	47.51 ± 0.06B	47.22 ± 0.13B
Glutamic acid ^1^	Umami (+)	117.25 ± 1.16c	118.41 ± 0.94c	119.75 ± 1.35bc	120.89 ± 0.99ab	122.45 ± 1.21a	122.11 ± 0.44A	121.81 ±0.21A	112.59 ± 0.11B	112.01 ± 2.10B	111.34 ± 0.31B
MSG-like FAA		167.02 ± 1.45c	168.66 ± 1.24c	170.55 ± 1.68bc	172.17 ± 1.33ab	174.38 ± 1.21a	173.89 ± 0.60A	173.47 ± 0.30A	160.35 ± 0.15B	159.52 ± 1.87B	158.56 ± 0.42B
Serine ^2^	Sweet (+)	19.20 ± 0.17c	19.37 ± 0.13c	19.57 ± 0.20bc	19.74 ± 0.15ab	19.97 ± 0.18a	19.92 ± 0.07A	19.87 ± 0.03A	18.38 ± 0.02B	18.28 ± 0.04B	18.17 ± 0.05B
Glycine ^2^	Sweet (+)	214.77 ± 2.15c	216.92 ± 1.94bc	219.40 ± 2.50abc	221.53 ± 1.83ab	224.43 ± 2.25a	223.79 ± 0.82A	223.24 ± 0.39A	206.32 ± 0.20B	205.26 ± 1.23B	204.05 ± 0.58B
Arginine ^2^	Bitter/sweet (+)	72.03 ± 0.70c	72.73 ± 0.43c	73.53 ± 0.80bc	74.21 ± 0.59 ab	75.15 ± 0.73a	74.94 ± 0.27A	74.77 ± 0.13A	69.12 ± 0.07B	68.76 ± 0.07B	68.35 ± 0.19B
Threonine ^2^	Sweet (+)	9.63 ± 0.09c	9.71 ± 0.11c	9.81 ± 0.10bc	9.90 ± 0.07ab	10.02 ± 0.09a	9.99 ± 0.03A	9.97 ± 0.01A	9.22 ± 0.01B	9.17 ± 0.03B	9.11 ± 0.03B
Alanine ^2^	Sweet (+)	83.66 ± 0.84d	84.49 ± 0.39cd	85.44 ± 0.96bc	86.26 ± 0.71 ab	87.39 ± 0.87a	87.14 ± 0.32A	86.92 ± 0.13A	80.34 ± 0.08B	79.93 ± 0.05B	79.45 ± 0.22B
Proline ^2^	Bitter/sweet (+)	49.44 ± 0.50c	49.93 ± 0.43bc	50.51 ± 0.58abc	51.00 ± 0.42ab	51.66 ± 0.52a	51.52 ± 0.19A	51.39 ± 0.02A	47.50 ± 0.05B	47.25 ± 0.03B	46.97 ± 0.13B
Sweet FAA		448.73 ± 3.99c	453.15 ± 3.35bc	458.26 ± 5.04abc	462.64 ± 3.07ab	468.62 ± 4.06a	467.30 ± 1.68A	466.16 ± 0.70A	430.88 ± 0.24B	428.65 ± 1.35B	426.10 ± 1.08B
Histidine ^3^	Bitter (−)	9.70 ± 0.07c	9.77 ± 0.08bc	9.85 ± 0.08abc	9.92 ± 0.06ab	10.02 ± 0.08a	10.00 ± 0.03A	9.98 ± 0.01A	9.24 ± 0.01B	9.19 ± 0.04B	9.13 ± 0.03B
Tyrosine ^3^	Bitter (−)	8.71 ± 0.04b	8.76 ± 0.10b	8.81 ± 0.05ab	8.85 ± 0.04 ab	8.91 ± 0.05a	8.90 ± 0.02A	8.89 ± 0.15A	8.25 ± 0.01B	8.19 ± 0.11B	8.14 ± 0.03B
Valine ^3^	Sweet/bitter (−)	10.25 ± 0.06bc	10.30 ± 0.07bc	10.37 ± 0.07abc	10.43 ± 0.05 ab	10.51 ± 0.02a	10.49 ± 0.02A	10.48 ± 0.09A	9.71 ± 0.02B	9.65 ± 0.07B	9.59 ± 0.03B
Methionine ^3^	Bitter/sweet/sulfurous (−)	4.35 ± 0.02b	4.37 ± 0.03b	4.40 ± 0.03ab	4.42 ± 0.02 ab	4.45 ± 0.04a	4.45 ± 0.01A	4.44 ± 0.00A	4.12 ± 0.01B	4.09 ± 0.03B	4.06 ± 0.01B
Cystine ^3^	Bitter/sweet/sulfurous (−)	4.91 ± 0.02a	4.93 ± 0.03a	4.95 ± 0.02a	4.96 ± 0.01a	4.98 ± 0.02a	4.98 ± 0.01B	4.98 ± 0.00B	4.98 ± 0.01B	4.99 ± 0.04B	5.00 ± 0.01A
Isoleucine ^3^	Bitter (−)	6.21 ± 0.04b	6.25 ± 0.06b	6.30 ± 0.05ab	6.34 ± 0.04 ab	6.40 ± 0.04a	6.39 ± 0.02A	6.38 ± 0.01A	5.91 ± 0.01B	5.87 ± 0.04B	5.83 ± 0.02B
Leucine ^3^	Bitter (−)	8.98 ± 0.06c	9.04 ± 0.11bc	9.11 ± 0.07abc	9.17 ± 0.05 ab	9.26 ± 0.07a	9.24 ± 0.02A	9.22 ± 0.01A	8.54 ± 0.01B	8.49 ± 0.05B	8.44 ± 0.03B
Phenylalanine ^3^	Bitter (−)	10.93 ± 0.06b	10.98 ± 0.13ab	11.05 ± 0.07ab	11.10 ± 0.05a	11.18 ± 0.06a	11.16 ± 0.02A	11.15 ± 0.01A	10.34 ± 0.02B	10.27 ± 0.04B	10.20 ± 0.04B
Tryptophan ^3^	Bitter (−)	10.32 ± 0.04b	10.37 ± 0.14ab	10.42 ± 0.05ab	10.46 ± 0.04a	10.52 ± 0.05a	10.50 ± 0.02A	10.49 ± 0.01A	9.74 ± 0.02B	9.68 ± 0.04B	9.61 ± 0.03B
Lysine ^3^	Sweet/bitter (−)	15.40 ± 0.11c	15.50 ± 0.09bc	15.63 ± 0.13abc	15.73 ± 0.09ab	15.88 ± 0.11a	15.85 ± 0.04A	15.82 ± 0.02A	14.65 ± 0.02B	14.57 ± 0.05B	14.47 ± 0.04B
Bitter FAA		89.76 ± 0.49c	90.27 ± 0.80bc	90.89 ± 0.97abc	91.38 ± 0.45ab	92.11 ± 0.51a	91.96 ± 0.21A	91.83 ± 0.30A	85.48 ± 0.14B	84.99 ± 0.48B	84.47 ± 0.26B
Asparagine ^4^	Flat/tasteless	10.79 ± 0.07b	10.86 ± 0.16b	10.94 ± 0.08ab	11.00 ± 0.06ab	11.10 ± 0.07a	11.08 ± 0.03A	11.06 ± 0.01A	10.25 ± 0.02A	10.19 ± 0.04AB	10.12 ± 0.03B
Glutamine ^4^	Flat/tasteless	53.94 ± 0.53d	54.47 ± 0.34cd	55.08 ± 0.61bc	55.60 ± 0.45ab	56.31 ± 0.55a	56.15 ± 0.20A	56.02 ± 0.10A	51.78 ± 0.05B	51.51 ± 0.13B	51.21 ± 0.15B
Tasteless FAA		64.73 ± 0.52d	65.33 ± 0.94cd	66.02 ± 0.66bc	66.60 ± 0.48ab	67.41 ± 0.61a	67.23 ± 0.22A	67.08 ± 0.11A	62.03 ± 0.06B	61.70 ± 0.16B	61.33 ± 0.18B
Total FAA		770.23 ± 7.19c	777.41 ± 6.61bc	785.71 ± 8.53abc	792.80 ± 6.11ab	802.52 ± 7.52a	800.37 ± 2.74A	798.53 ± 1.32A	738.75 ± 0.78B	734.86 ± 3.76B	730.46 ± 2.05B

Data are mean ± standard deviation (n = 3). Values in rows with different lowercase and uppercase letters represent significant differences in the ammonia and salinity treatments, respectively (*p* < 0.05). ^1^ MSG-like FAAs; ^2^ sweet FAAs; ^3^ bitter FAAs; ^4^ tasteless FAAs.

**Table 2 foods-12-03200-t002:** The concentrations of nucleotides (mg/100 g wet weight) in the ivory shell meat under different ammonia and salinity levels.

Item	Ammonia Concentrations (mg/L)					Salinity (ppt)				
	0	2.5	5	7.5	10	20	25	30	35	40
CMP	ND	ND	ND	ND	ND	ND	ND	ND	ND	ND
UMP	3.75 ± 0.57a	3.54 ± 0.38ab	2.21 ± 0.05b	2.12 ± 0.07b	4.83 ± 0.96a	1.53 ± 0.19AB	1.53 ± 0.26AB	2.12 ± 0.09A	1.49 ± 0.20B	1.66 ± 0.65AB
GMP	ND	ND	ND	ND	ND	ND	ND	ND	ND	ND
IMP	ND	ND	ND	ND	ND	ND	ND	ND	ND	ND
AMP	129.90 ± 16.37c	180.96 ± 12.11a	155.01 ± 0.13b	155.93 ± 1.37b	180.83 ± 5.34a	140.41 ± 12.46AB	159.66 ± 8.53A	133.87 ± 8.15B	148.14 ± 17.03AB	146.30 ± 12.72AB
Total	133.64 ± 17.94c	184.50 ± 11.73a	157.22 ± 0.07b	158.05 ± 1.30b	186.66 ± 6.29a	141.95 ± 12.66AB	161.19 ± 8.80A	135.99 ± 8.06B	149.63 ± 16.83AB	147.96 ± 12.07AB

Data are mean ± standard deviation (n = 3). Values in rows with different lowercase and uppercase letters represent significant differences in the ammonia and salinity treatments, respectively (*p* < 0.05). ND, not detected.

**Table 3 foods-12-03200-t003:** The concentrations of organic acids (mg/g wet weight) in the ivory shell meat under different ammonia and salinity levels.

Item	Ammonia Concentrations (mg/L)					Salinity (ppt)				
	0	2.5	5	7.5	10	20	25	30	35	40
Lactic acid	2.56 ± 0.29b	2.33 ± 0.01b	2.27 ± 0.31b	3.74 ± 0.48a	2.08 ± 0.26b	2.39 ± 0.16A	2.31 ± 0.01A	2.57 ± 0.10A	2.06 ± 0.50A	2.21 ± 0.42A
Acetic acid	5.43 ± 0.87a	4.50 ± 0.19a	4.63 ± 0.25a	4.45 ± 0.19a	5.10 ± 0.42a	4.28 ± 1.06AB	4.13 ± 1.05AB	3.28 ± 0.24BC	4.69 ± 0.26A	2.63 ± 0.39C
Citric acid	26.07 ± 2.69d	31.44 ± 0.68c	35.94 ± 2.90ab	34.28 ± 0.52bc	39.00 ± 1.53a	32.60 ± 1.2A	29.57 ± 0.25B	26.54 ± 0.53C	30.12 ± 0.98B	28.84 ± 1.14B
Malic acid	ND	ND	ND	ND	ND	ND	ND	ND	ND	ND
Succinic acid	ND	ND	ND	ND	ND	ND	ND	ND	ND	ND

Data are mean ± standard deviation (n = 3). Values in rows with different lowercase and uppercase letters represent significant differences in the ammonia and salinity treatments, respectively (*p* < 0.05). ND, not detected.

**Table 4 foods-12-03200-t004:** The concentrations of mineral ions (wet weight) in the ivory shell meat under different ammonia and salinity levels.

Item	Ammonia Concentrations (mg/L)					Salinity (ppt)				
	0	2.5	5	7.5	10	20	25	30	35	40
Ca^2+^ (mg/g)	3.00 ± 0.09c	4.27 ± 0.06a	3.04 ± 0.06c	3.41 ± 0.03b	2.98 ± 0.18c	2.70 ± 0.11D	3.66 ± 0.36A	3.28 ± 0.04BC	3.46 ± 0.09AB	3.00 ± 0.07CD
Na^+^ (mg/g)	4.77 ± 0.08b	4.81 ± 0.10ab	4.77 ± 0.04b	4.92 ± 0.04ab	5.17 ± 0.35a	3.97 ± 0.03E	4.31 ± 0.09D	5.04 ± 0.12C	5.58 ± 0.11B	7.06 ± 0.06A
K^+^ (mg/g)	3.64 ± 0.02a	3.58 ± 0.03a	3.65 ± 0.03a	3.66 ± 0.01a	3.59 ± 0.09a	3.59 ± 0.11C	3.76 ± 0.04B	3.69 ± 0.04BC	3.48 ± 0.04D	4.02 ± 0.04A
Mg^2+^ (mg/g)	3.99 ± 0.09a	3.72 ± 0.04cd	3.96 ± 0.22ab	3.81 ± 0.03bc	3.58 ± 0.06d	3.67 ± 0.28B	3.69 ± 0.07B	3.89 ± 0.09B	4.05 ± 0.03B	4.94 ± 0.33A
Zn^2+^ (mg/kg)	16.88 ± 0.63c	15.45 ± 0.13d	27.37 ± 0.40a	24.01 ± 0.06b	16.76 ± 0.04c	16.87 ± 0.37C	28.19 ± 0.63A	16.11 ± 0.27CD	15.93 ± 0.59D	21.13 ± 0.38B
Mn^2+^ (mg/kg)	1.34 ± 0.05c	1.53 ± 0.06ab	1.50 ± 0.03b	1.62 ± 0.05a	1.36 ± 0.09c	1.15 ± 0.02D	1.52 ± 0.01A	1.34 ± 0.00C	1.40 ± 0.06B	1.56 ± 0.00A
Cl^−^ (mg/g)	1.25 ± 0.05c	1.31 ± 0.01b	1.33 ± 0.00b	1.30 ± 0.01b	1.43 ± 0.01a	1.35 ± 0.03B	1.28 ± 0.04B	1.38 ± 0.07B	1.45 ± 0.06B	1.72 ± 0.20A
PO_4_^3−^ (mg/g)	2.59 ± 0.24a	2.64 ± 0.17a	2.79 ± 0.36a	2.70 ± 0.00a	2.96 ± 0.08a	3.02 ± 0.25B	3.26 ± 0.12AB	3.25 ± 0.10AB	3.11 ± 0.15B	3.54 ± 0.37A

Data are mean ± standard deviation (n = 3). Values in rows with different lowercase and uppercase letters represent significant differences in the ammonia and salinity treatments, respectively (*p* < 0.05).

## Data Availability

The datasets generated for this study are available on request to the corresponding author.

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
