# Peer review of "Effects of Ammonia and Salinity Stress on Non-Volatile and Volatile Compounds of Ivory Shell (Babylonia areolata)"

_foods, 2023, doi:10.3390/foods12173200_

Round 1
Reviewer 1 Report
Dear Authors
Your manuscript is in good scientific form and your English language is good. I set some comments for enhanced the manuscript before further decision:
Line 26: change "its" to "it's"
Line 73: set exact treatment number instead of "different".
Line 104-105: add the information of the column, standard and the run conditions.
Line 109-110: add the change concentration slope of two different eluent and run time.
Line 112: define the MSG in first appearance.
Line 126: change 0.45-µm to 0.45 µm
Line 144: "30 m" change to "30m"
Line 164- 166: rewrite the sentence
Line 257: Correct to Cl- in
Line 259: Correct to Mn2+
Line 270-276: add references
Table 3: you write units in different styles. for example in table 1 mg/100 g but in other tables mg.L-1
it is better to unify the reported unit in the whole manuscript body
Figure 1: the resolution of the figure must be completely enhanced and the graph must directly transfer from excel. Also, A and B set out of the figures.
Line 369: correct to Cl- in
Conclusion: add comments for rearing pounds about better taste production based on ammonia and salinity regulating.
Author Response
Dear reviewer,
Thanks for your comments. We have studied comments carefully and have made correction which we hope to meet with approve. Revised portions are marked in red in the manuscript. Our response of the comments is enclosed at the end of this letter. Thank you again for your assistance.
- Line 26: change "its" to "it's"
Response to comments: The mistake has been corrected.
- Line 73: set exact treatment number instead of "different".
Response to comments: According to the reviewer’s advice, the exact treatment number has been added.
- Line 104-105: add the information of the column, standard and the run conditions.
Response to comments: Thanks. More information about the FAA analysis has been provided in the manuscript. In detail, “samples (5 µL) were separated using a Waters Pico-Tag-C18 column (3.9 mm × 150 mm). The mobile phase consisted of AccQ-Tag Elent A (v/v = 10:1) (A), 100% acetonitrile (B) and deionized water (C). Gradient elution was performed as follows: 100% A to 99% A and 1% B at 0.5 min, to 94% A and 6% B at 18 min, to 90% A and 10% B at 19 min, to 83% A and 17% B at 29.5 min, to 60% B and 40% C at 33 min, to 100% A at 36 min, and to 100% A at 45 min. The detection wavelength was 248 nm. All analyses were repeated in triplicate.”
- Line 109-110: add the change concentration slope of two different eluent and run time.
Response to comments: The concentration changes of eluent A and B have been added in the manuscript. In detail, “the following gradient was performed: initial of 5 % A for 10 min, linear change to 15 % A for 5 min, linear change to 70 % A for 6 min and finally linear change to 5 % A for 4 min”.
- Line 112: define the MSG in first appearance.
Response to comments: The full name of MSG has been added in abstract and section 2.5 parts.
- Line 126: change 0.45-µm to 0.45 µm
Response to comments: The mistake has been corrected.
- Line 144: "30 m" change to "30m"
Response to comments: Thanks. The mistake has been corrected.
- Line 164- 166: rewrite the sentence
Response to comments: The sentence has been rewritten as follow: “For example, the top three FAAs were Ala, Gly and Glu in Hongkong oyster (Crassostrea hongkongensis), which were almost the same as our study [24], while Gly, Ala, histidine (His) in Suminoe oyster (Crassostrea ariakensis) [14]; for mud crab, the mainly FAAs were proline (Pro), arginine (Arg) and Gly [25]; for Pacific white shrimp, they were lysine (Lys), leucine (Leu) and Arg in [26]; and for grouper, they were Gly, Ala and Lys [15].”
- Line 257: Correct to Cl-in; Line 259: Correct to Mn2+
Response to comments: These formatting errors has been corrected.
- Line 270-276: add references
Response to comments: Thanks. The references have been added in the manuscript.
- Table 3: you write units in different styles. for example in table 1 mg/100 g but in other tables mg.L-1. it is better to unify the reported unit in the whole manuscript body.
Response to comments: Thanks. In this manuscript, the differences of unit of detected flavor compounds are intended to make the data more readable. For example, the data of citric acid contents will be 2607-3928 mg/100 g of wet weight when the units were unifying to mg/100 g. Therefore, different units were used among different flavor compounds.
- Figure 1: the resolution of the figure must be completely enhanced and the graph must directly transfer from excel. Also, A and B set out of the figures.
Response to comments: The figure has been changed accordingly.
- Line 369: correct to Cl-in
Response to comments: The mistake has been corrected.
- Conclusion: add comments for rearing pounds about better taste production based on ammonia and salinity regulating.
Response to comments: According to the reviewer’s advice, the comments has been added in the end of conclusion part. In detail, “In conclusion, slight ammonia elevation (2.5 mg/L) and low salinity (25 ppt) could improve the characteristic taste of ivory shell.”
Reviewer 2 Report
Effects of ammonia and salinity stress on non-volatile and volatile compounds of ivory shell (Babylonia areolata)
1. Line 50-68. More explanation is required for the mechanism underlying the effect of salinity and ammonia concentration on seafood flavor characteristics. Also, do they have an impact on the muddy odor of the seafood? Please cite some publications about the muddy/earthy odor in aquatic foods below.
Phetsang, H., Panpipat, W., Undeland, I., Panya, A., Phonsatta, N., & Chaijan, M. (2021). Comparative quality and volatilomic characterisation of unwashed mince, surimi, and pH-shift-processed protein isolates from farm-raised hybrid catfish (Clarias macrocephalus× Clarias gariepinus). Food chemistry, 364, 130365.
Phetsang, H., Panpipat, W., Panya, A., Phonsatta, N., Cheong, L. Z., & Chaijan, M. (2022). Chemical characteristics and volatile compounds profiles in different muscle part of the farmed hybrid catfish (Clarias macrocephalus× Clarias gariepinus). International Journal of Food Science & Technology, 57(1), 310-322.
2. Line 91-92. Why only 90 ivory shells?
3. Line 90-96. How about the growth performance of the shells?
4. Line 97-98. Please provide further information on how to obtain the muscle. How were the shells killed? How long will the frozen storage at -80°C last? These characteristics will undoubtedly influence the flavor profile of the sample.
5. In the analyses, what about the off-flavor/off-odor compounds influenced by the treatments, i.e. bitter substances?
6. How about the flavor of the cooked sample to make it more realistic? Is this shell eaten raw?
7. Is there a relationship between the glycine content and the collagen content in this sample?
8. How about the pH of the muscle under different conditions? Also, how about the total volatile base N content in the muscle of each treatment?
9. Line 234-252. It is necessary to indicate the sample's freshness herein since it can alter glycolysis and, as a result, ATP depletion, which will affect both nucleotides and organic acids, as well as other metabolites of the samples.
10. The synergistic impact of nucleotide, amino acid, and salt on umami taste should be discussed.
11. Section 3.4. The fatty acid profile of the samples is required to correlate with flavor profiles, particularly those produced from lipid oxidation/degradation.
12. Not only flavor, but also texture, water holding capacity, color, and cooking loss of the muscle should be considered. Also, post-harvest flavor changes, as well as the effect of cooking methods, might influence such flavors. It should be suggested in the conclusion regarding this task that needs to be completed later.
Minor editing of English language required
Author Response
Dear reviewer,
Thanks for your comments. We have studied comments carefully and have made correction which we hope to meet with approve. Revised portions are marked in red in the manuscript. Our response of the comments is enclosed at the end of this letter. Thank you again for your assistance.
- Line 50-68. More explanation is required for the mechanism underlying the effect of salinity and ammonia concentration on seafood flavor characteristics. Also, do they have an impact on the muddy odor of the seafood? Please cite some publications about the muddy/earthy odor in aquatic foods below.
Phetsang, H., Panpipat, W., Undeland, I., Panya, A., Phonsatta, N., & Chaijan, M. (2021). Comparative quality and volatilomic characterisation of unwashed mince, surimi, and pH-shift-processed protein isolates from farm-raised hybrid catfish (Clarias macrocephalus× Clarias gariepinus). Food chemistry, 364, 130365.
Phetsang, H., Panpipat, W., Panya, A., Phonsatta, N., Cheong, L. Z., & Chaijan, M. (2022). Chemical characteristics and volatile compounds profiles in different muscle part of the farmed hybrid catfish (Clarias macrocephalus× Clarias gariepinus). International Journal of Food Science & Technology, 57(1), 310-322.
Response to comments: According to the reviewer’s advice, the mechanism of salinity and ammonia concentration on seafood flavor characteristics has been added in the introduction parts. Furthermore, the differences of muddy odor between freshwater fish and marine fish are also added, and new references (including the two papers provided by the reviewer) are cited. In detail, “In previous studies, environmental changes were shown to have an impact on physiological metabolism (such as glycolysis, lipolysis, respiratory metabolism, etc.) of aquatic animals, leading to the changes of non-volatile and volatile compounds [11, 13, 16]. For example, razor clams cultured at high salinity (20-25 ppt) possessed more positive, abundant volatiles than those cultured at low salinity [9]; low salinity decreased the content of flavor compounds and weakened the umami taste and aroma characteristics of mud crab (Scylla paramamosain) [17]; high ammonia concentrations could reduce umami amino acids and increase bitter amino acid levels in Chinese mitten crab meat [11]; freshwater fish (such as catfish) had stronger muddy odour compared to marine fish, which was mainly casued by the higher contents of 2-methylisoborneol, o (E)-2-nonenal, 1-octen-3-ol and other earthy-musty compounds in freshwater fish [18, 119].”
- Line 91-92. Why only 90 ivory shells?
Response to comments: Thanks. It was 90 ivory shells for each ammonia or salinity treated group. In order to this eliminate ambiguous meaning, this part has been rewritten as follow: “Based on our previous studies [2, 4], five ammonia concentrations (0, 2.5, 5.0, 7.5, and 10.0 mg/L) and salinity levels (20, 25, 30, 35, and 40 ppt) of long-term stress were set. In ammonia stress experiments, ammonium chloride (NH4Cl, Sinopharm Chemical Reagent Co., Ltd., Shanghai, China) was added to achieve the desired final ammonia concentration. All ammonia-treated groups were conducted in triplicate. For each group, a total of 90 ivory shells were randomly divided into three sub-groups and cultured in three 68-L blue tanks (30 ivory shells for each tank).”
- Line 90-96. How about the growth performance of the shells?
Response to comments: Thanks. The effects of ammonia and salinity stress on growth performance of ivory shells has been performed and published in previous studies (Liu et al., 2023; Zhou et al., 2023). The results showed that (1) the growth performance of ivory shells in the higher ammonia treatments (≥ 5.0 mg/L) was significantly lower than that of the control group; (2) the growth performance of ivory shells in the 20-, 25- and 30-ppt groups were significantly higher than those cultured at higher salinity (35 and 40 ppt).
References:
- Liu, C.; Zhou, J.; Yang, Y.; Yang, Y.; Wang, A.; Gu, Z. Effects of salinity on growth performance, physiological response, and body biochemical composition of juvenile ivory shell (Babylonia areolata). Aquaculture 2023, 566, 739193.
- Zhou, J.; Liu, C.; Yang, Y.; Yang, Y.; Gu, Z.; Wang, A.; Liu, C. Effects of long-term exposure to ammonia on growth performance, immune response, and body biochemical composition of juvenile ivory shell, Babylonia areolate. Aquaculture 2023, 562, 738857.
- Line 97-98. Please provide further information on how to obtain the muscle. How were the shells killed? How long will the frozen storage at -80°C last? These characteristics will undoubtedly influence the flavor profile of the sample.
Response to comments: According to the reviewer’s advice, more information about the ivory shell sampling and muscle storage are added in the manuscript. In detail, “the ivory shells were sacrificed 24 h after the last feeding. The muscle of ivory shells in each tank was sampled, quickly frozen with liquid nitrogen and store at −80 °C for non-volatile and volatile compounds analyses. All chemical analyses were performed within two months.”
- In the analyses, what about the off-flavor/off-odor compounds influenced by the treatments, i.e. bitter substances?
Response to comments: In this manuscript, the total contents of MSG-like FAAs, sweet FAAs, bitter FAAs and tasteless FAAs are calculated and added in Table 1. Accordingly, the results of these four tastes FAAs have been added in Line 198-201 [The contents of MSG-like and sweet FAAs in ivory shells were 158.56-174.38 and 426.10-468.62 mg per 100 g of wet weight, respectively, which showed significantly higher than bitter FAA (84.47-92.11 mg per 100 g of wet weight)]. However, in ivory shell, only four FAAs (Glu, Gly, Arg and Ala) significantly contributed to taste (taste activity values (TAVs) were > 1). Therefore, the bitter FAAs were not discussed in our manuscript.
- How about the flavor of the cooked sample to make it more realistic? Is this shell eaten raw?
Response to comments: Yes. The ivory shells are eaten by boiling or baking with salt in China. In this study, we focus on the changes of non-volatile and volatile compounds of ivory shell cultured under different ammonia and salinity levels. Therefore, we did not perform special processes (such as boiling or baking with salt) before flavor compounds analyses. In fact, the flavor compounds analyses of raw seafood were also observed in many other species. Anyway, it is a good idea that we might analyze the flavor compounds of ivory shell after cooking in the following-up experiment.
- Is there a relationship between the glycine content and the collagen content in this sample?
Response to comments: Glycine is a major quantitative component of collagen. Until now, collagen has been extracted and studied from different marine invertebrates, such as abalone, cuttlefish, squid, jellyfish, and starfish. However, as far as our known, there are no studies on the relationship between glycine content and the collagen content in ivory shell.
- How about the pH of the muscle under different conditions? Also, how about the total volatile base N content in the muscle of each treatment?
Response to comments: Thanks. As mentioned in Question 12, not only pH, but also texture, water holding capacity, color, and cooking loss of the muscle are important parameters to evaluate the quality of food. We are sorry that we do not focus on these quality properties in this manuscript, which might be performed in follow-up experiment.
It has been reported that the content of total volatile base N is generally used as a chemical index for fish freshness and fishy odor. In our experiment, “the muscle of ivory shell was sampled, quickly frozen with liquid nitrogen and store at −80 °C. All chemical analyses were performed within two months.” The detected muscle was fresh. Therefore, the total volatile base N contents among different groups were not performed.
- Line 234-252. It is necessary to indicate the sample's freshness herein since it can alter glycolysis and, as a result, ATP depletion, which will affect both nucleotides and organic acids, as well as other metabolites of the samples.
Response to comments: According to the reviewer’s advice. The relationship between freshness and organic acids of ivory shell muscle has been added in Line 271-275. In detail, “It has been reported that the sample’s freshness also alters the content of organic acids by glycolysis. In this study, the ivory shell muscle was well preserved (stored at -80 °C for no more than two months). Therefore, the differences of organic acid contents of different ivory shell groups were not caused by storage condition.”
- The synergistic impact of nucleotide, amino acid, and salt on umami taste should be discussed.
Response to comments: Thanks. As shown in our manuscript, the synergistic impact of nucleotide, amino acid, and inorganic ions on umami taste has been discussed in Line 322-340 “section 3.3. Effect of ammonia and salinity on EUC of ivory shells”. In this part, the EUC values determined by the mixture of umami amino acids and 5′-nucleotides were calculated. The enhancement (Na+, Cl− and PO43−) and inhibiting effect (Ca2+) of different inorganic ions on umami taste were also discussed.
- Section 3.4. The fatty acid profile of the samples is required to correlate with flavor profiles, particularly those produced from lipid oxidation/degradation.
Response to comments: As described by the reviewer, there are close relationships between fatty acid profile, particularly lipid oxidation/degradation, and flavor profiles. In our previous studies, the fatty acid profiles of ivory shell muscle of different ammonia and salinity treated groups have been studied (Liu et al., 2023; Zhou et al., 2023). In this manuscript, the relationship of volatiles and fatty acid profile in muscle was discussed in Line 368-381 (third paragraph of Section 3.4). In detail, “Production of volatiles in seafood depends on lipid auto-oxidation and enzymatic oxidation [40]. Aldehydes are mainly formed from lipid oxidation, and most of them have a fatty and fruity aroma [43]. As the major aldehyde, hexanal is a derivative of linoleic acid and arachidonic acid oxidation, which has an odor described as oxidized fatty, green, grassy, powerful, and penetrating [9]. In our previous study, only linoleic acid was detected in the ivory shell, and its profile in the total lipid of muscle showed the opposite tendency with hexanal when the ivory shell suffers the same ammonia and salinity stresses [2, 4]. We indicated that hexanal in ivory shell were probably produced from linoleic acid oxidation. Acids are formed through the oxidation of aldehydes and have cheesy ammoniacal odors [44]. In ivory shell, as the butanoic acid, 4-(dimethylamino)-3-hydroxy- accounting for 6.64%-14.60% of total detected volatile components, this acid might have a great effect on the favor of ivory shell. Ketones are produced via microbially induced oxidation, lipid oxidation or amino acid degradation and possess a heavy rose fragrance [45].”
- Not only flavor, but also texture, water holding capacity, color, and cooking loss of the muscle should be considered. Also, post-harvest flavor changes, as well as the effect of cooking methods, might influence such flavors. It should be suggested in the conclusion regarding this task that needs to be completed later.
Response to comments: Thanks. According to the reviewer’s advice, this corresponding content has been added in the conclusion part. In detail, “In addition to non-volatile and volatile compounds, other quality indexes, such as texture, water holding capacity, color, cooking loss, and post-harvest flavor changes of the muscle should be detected in follow-up study.”
Reviewer 3 Report
Review report “foods-2540332”
In the manuscript entitled “Effects of ammonia and salinity stress on non-volatile and volatile compounds of ivory shell (Babylonia areolata)” the authors report on the non-volatile and volatile compounds of ivory shell and their changes caused by ammonia and salinity stresses.
The manuscript is well written, organized in a logical way and deals with the subject in great detail. The length is appropriate. The abstract is quite clear and sufficiently reflects the manuscript content.
Although the topic is really specific and its interest may be limited, it has some merits.
I have some suggestion and comments for the authors:
M&M This section should be implemented.
- Add (Company, City, Country) for all instrumentations reagents standards, software. Standardize
and change it in all manuscript (e.g., lines 90,102, 109,110, 125,126,134,138 etc)
- line 90-96 this part should be better explained. It is not clear in how many groups you divided the samples (five I guess from the tables). And for ammonia stress experiments each level (0, 2.5, 5.0, 7.0 and 10.0 mg L-1) was replicated three times. Is it right?
- SPME: explain
- How many replicates for each analysis? (Paragraphs: 2.3, 2.4, 2.6, 2.7, 2.8)
- Use g/L instead of ppt, change it.
- Units: use the same format, choose if you prefer “u/u” or “u u-1”, and use it in all the manuscript.
- Along the manuscript the authors repeat excessively “in this study”, “in our study”, “our study”. It may result a bit boring during the reading.
I suggest the authors add a paragraph before the conclusions in which talk about the possible applications of their findings for the ivory shell aquaculture.
Based on these comments I strongly encourage the authors to improve the manuscript, since it may be a good candidate for publication in Foods after the revisions.
Author Response
Dear reviewer,
Thanks for your comments. We have studied comments carefully and have made correction which we hope to meet with approve. Revised portions are marked in red in the manuscript. Our response of the comments is enclosed at the end of this letter. Thank you again for your assistance.
- Add (Company, City, Country) for all instrumentations reagents standards, software. Standardize
and change it in all manuscript (e.g., lines 90,102, 109,110, 125,126,134,138 etc)
Response to comments: Thanks. All the information has been added in the manuscript.
- line 90-96 this part should be better explained. It is not clear in how many groups you divided the samples (five I guess from the tables). And for ammonia stress experiments each level (0, 2.5, 5.0, 7.0 and 10.0 mg L-1) was replicated three times. Is it right?
Response to comments: Yes. The group numbers of ammonia and salinity level are both five, and three replications were performed in each treatment. Furthermore, this part has been rewritten as follow, “Based on our previous studies [2, 4], five ammonia concentrations (0, 2.5, 5.0, 7.5, and 10.0 mg/L) and salinity levels (20, 25, 30, 35, and 40 ppt) of long-term stress were set. In ammonia stress experiments, ammonium chloride (NH4Cl, Sinopharm Chemical Reagent Co., Ltd., Shanghai, China) was added to achieve the desired final ammonia concentration. All ammonia-treated groups were conducted in triplicate. For each group, a total of 90 ivory shells were randomly divided into three sub-groups, and cultured in three 68-L blue tanks (30 ivory shells for each tank). The cultured condition of each ammonia stress group was the same as temporary rearing experiment, except that water were changed with the same ammonia concentration before. For salinity stress experiments, the experimental condition was as same as above, and the water salinity of each salinity-treated group was maintained by adding commercial sea salt or purified freshwater to the natural seawater (~31.2 ppt).”
- SPME: explain
Response to comments: The full name (solid-phase microextraction) has been added.
- How many replicates for each analysis? (Paragraphs: 2.3, 2.4, 2.6, 2.7, 2.8)
Response to comments: In this experiment, three replicates of each analysis were performed. The information has been added in the manuscript.
- Use g/L instead of ppt, change it.
Response to comments: Thanks. Though g/L and ppt are the same salinity unit. However, the unit of ppt is more widely used (such as in Resley et al., 2006; Mohamed et al., 2021, and Liu et al., 2023) in aquaculture relative papers when compared to g/L. Therefore, the salinity unit “ppt” is still retained in this manuscript.
References:
- Resley, M.J.; Webb, K.A. J.; Holt, G. J.. Growth and survival of juvenile cobia, Rachycentron canadum, at different salinities in a recirculating aquaculture system. Aquaculture 2006, 253(1–4), 398–407.
- Mohamed, N.A.; Saad, M.F.; Shukry, M.; El-Keredy, A.M.S.; Nasif, O.; Doan, H.V.; Dawood, M.A.O. Physiological and ion changes of Nile tilapia (Oreochromis niloticus) under the effect of salinity stress. Aquacult. Rep. 2021, 19, 100567.
- Liu, C.; Zhou, J.; Yang, Y.; Yang, Y.; Wang, A.; Gu, Z. Effects of salinity on growth performance, physiological response, and body biochemical composition of juvenile ivory shell (Babylonia areolata). Aquaculture 2023, 566, 739193.
- Units: use the same format, choose if you prefer “u/u” or “u u-1”, and use it in all the manuscript.
Response to comments: Thanks. The format of the units has been unified.
- Along the manuscript the authors repeat excessively “in this study”, “in our study”, “our study”. It may result a bit boring during the reading.
Response to comments: According to the reviewer’s advice, most of these repetitive words have been deleted.
- I suggest the authors add a paragraph before the conclusions in which talk about the possible applications of their findings for the ivory shell aquaculture.
Response to comments: Thanks. Based on the comments of three reviewers, more quality indexes (that needs to be completed in follow-up study) and comments for rearing ivory shells were added at the end of the conclusion. In detail, “In addition to non-volatile and volatile compounds, other quality indexes, such as texture, water holding capacity, color, cooking loss, and post-harvest flavor changes of the muscle should be detected in follow-up study. In conclusion, slight ammonia elevation (2.5 mg/L) and low salinity (25 ppt) could improve the characteristic taste of ivory shell.”
Round 2
Reviewer 2 Report
All reviewer concerns were thoroughly addressed and handled point by point. As a result, it is acceptable in the presence form.
-
Reviewer 3 Report
Ok.
Some minor changes/typos:
- line 96 formula
- line 112 space
- line 115 add particle size
- line 116 for the three mobile phases add analytical grade and Company, City, Country
Moderate editing of English language is required